# Distinctive Tissue and Serum MicroRNA Profile of IgG4-Related Ophthalmic Disease and MALT Lymphoma

**DOI:** 10.3390/jcm9082530

**Published:** 2020-08-05

**Authors:** Naoya Nezu, Yoshihiko Usui, Masaki Asakage, Hiroyuki Shimizu, Kinya Tsubota, Akitomo Narimatsu, Kazuhiko Umazume, Naoyuki Yamakawa, Shin-ichiro Ohno, Masakatsu Takanashi, Masahiko Kuroda, Hiroshi Goto

**Affiliations:** 1Department of Ophthalmology, Tokyo Medical University, Tokyo 160-0023, Japan; naoya.nezu@gmail.com (N.N.); patty.m.best@gmail.com (M.A.); sardine_harbor@yahoo.co.jp (H.S.); tsubnkin@hotmail.co.jp (K.T.); a.narimatsu1128@gmail.com (A.N.); kazuhiko-uma@kvf.biglobe.ne.jp (K.U.); yamakawa@tokyo-med.ac.jp (N.Y.); goto1115@tokyo-med.ac.jp (H.G.); 2Department of Molecular Pathology, Tokyo Medical University, Tokyo 160-8402, Japan; s-ohno@tokyo-med.ac.jp (S.-i.O.); m-takana@tokyo-med.ac.jp (M.T.); kuroda@tokyo-med.ac.jp (M.K.)

**Keywords:** microRNA, orbital lymphoproliferative disorders, IgG4-related disease, IgG4-related ophthalmic disease, orbital MALT lymphoma, machine-learning

## Abstract

The molecular pathogenesis of orbital lymphoproliferative disorders, such as immunoglobulin G4-related ophthalmic disease (IgG4-ROD) and orbital mucosa-associated lymphoid tissue (MALT) lymphoma, remains essentially unknown. Differentiation between the two disorders, which is important since the work-up and treatment can vary greatly, is often challenging due to the lack of specific biomarkers. Although miRNAs play an important role in the regulation of carcinogenesis and inflammation, the relationship between miRNA and orbital lymphoproliferative diseases remains unknown. We performed a comprehensive analysis of 2565 miRNAs from biopsy and serum specimens of 17 cases with IgG4-ROD, where 21 cases with orbital MALT lymphoma were performed. We identified specific miRNA signatures and their miRNA target pathways, as well as the network analysis for IgG4-ROD and orbital MALT lymphoma. Machine-learning analysis identified miR-202-3p and miR-7112-3p as the best discriminators of IgG4-ROD and orbital MALT lymphoma, respectively. Enrichment analyses of biological pathways showed that the longevity-regulating pathway in IgG4-ROD and the mitogen-activated protein kinase (MAPK) signaling pathway in orbital MALT lymphoma was most enriched by target genes of downregulated miRNAs. This is the first evidence of miRNA profiles of biopsy and serum specimens of patients with IgG4-ROD and orbital MALT lymphoma. These data will be useful for developing diagnostic and therapeutic interventions, as well as elucidating the pathogenesis of these disorders.

## 1. Introduction

Orbital lymphoproliferative tumors are composed of a heterogeneous group that includes malignant lymphomas, such as mucosa-associated lymphoid tissue (MALT) lymphoma, follicular lymphoma, and diffuse large B-cell lymphoma, as well as benign lymphoproliferative tumors, such as immunoglobulin G4-related ophthalmic disease (IgG4-ROD) and reactive lymphoid hyperplasia. It is difficult to differentiate malignant lymphoma from benign lymphoproliferative disorders, as these lesions share common clinical symptoms, imaging diagnosis, and histologic features, as well as molecular markers [1,2,3,4,5,6]. Clinical symptoms, imaging diagnosis, histology, molecular analysis, and flow cytometry using biopsy specimens are the most commonly used approaches for the diagnosis of orbital lymphoproliferative disorders. In particular, the kappa/lambda ratio of a biopsy specimen is useful for verifying the monoclonality of malignant B cells. However, the wide range of kappa/lambda or lambda/kappa ratios (1.4–6.0) limits its specificity, and that test is not useful in identifying and distinguishing certain cases of orbital MALT lymphoma [7,8,9]. Additionally, Sato et al. [10] detected IgG heavy-chain gene rearrangements in two of 17 cases of IgG4-ROD, further increasing the challenge in differentiating IgG4-ROD from orbital MALT lymphoma without specific analysis. Furthermore, several reports have indicated that orbital MALT lymphoma [10,11] and diffuse large B-cell lymphoma [12] develop based on IgG4-ROD, suggesting that these disorders are biologically related.

Therefore, correct diagnosis of IgG4-ROD and orbital MALT lymphoma (where both are of B-cell lineage) in routine clinical practice remains complicated and challenging due to the lack of specific diagnostic biomarkers. Elevated serum IgG4 is not sufficiently sensitive or specific for this purpose [13,14]. Therefore, approximately one-third of patients with IgG4-ROD do not meet the clinical criteria, and are diagnosed as “possible” or “probable” IgG4-ROD, which contributes to clinical confusion and delayed diagnosis [15,16].

MicroRNAs (miRNAs) are small non-coding RNA molecules (approximately 22 nucleotides in length) involved in the regulation of gene expression by partial base-pairing of target messenger RNAs (mRNA) to complementary sequences, which leads to cleavage and eventual degradation of the target mRNA or translational repression [17,18]. Recently, the role of non-coding RNAs, of which miRNAs are the most studied, has acquired remarkable importance in the pathogenesis of many types of disease, including cancer and inflammatory diseases. miRNAs have shown promise in recent years as biomarkers for the diagnosis and prognosis of cancer because of their diverse but tissue- and cell-specific biological and pathological functions [19]. Currently, more than 2500 miRNAs have been identified in human genomes. However, there is limited knowledge regarding the expression of miRNA. Moreover, no high-throughput miRNA expression studies have been conducted to identify miRNAs specifically associated with diseases, and no biopsy tissue and serum miRNA profiling of orbital lymphoproliferative disorders has been performed to date.

In this study, we performed a comprehensive miRNA microarray analysis using the 3D-Gene Human miRNA Oligo Chip, aiming to identify differentially expressed miRNAs and pathways using biopsy and serum specimens from patients with orbital MALT lymphoma and IgG4-ROD using machine-learning methods. Through the analysis of the targeted miRNA data, we investigated their potential implication in disease pathogenesis and their usefulness as biomarkers.

## 2. Materials and Methods

### 2.1. Patients

The diagnosis of IgG4-ROD and orbital MALT lymphoma was based on clinical, radiographic, histologic, flow cytometric, and molecular genetic (e.g., gene rearrangement) analyses of biopsy specimens. In particular, the diagnosis of IgG4-ROD was made according to published criteria [15]. Results of laboratory examinations, including serum IgG4, were recorded. Patients did not receive systemic corticosteroids, radiation therapy, or chemotherapy before sampling.

Written informed consent was provided by all participants in the study. The study was approved by the local Ethical Committee of the Tokyo Medical University Hospital, Tokyo, Japan (2016-162). All investigations were conducted in accordance with the principles of the Declaration of Helsinki.

### 2.2. Blood and Biopsy Sample Collection

Venous blood samples (approximately 5.0 mL) were collected in BD vacutainer tubes using a 21-gauge needle from patients and controls enrolled in this study. The samples were centrifuged (8000 rpm, 4 °C, 15 min) to collect the serum, which was stored at −80 °C until further analysis. Biopsy specimens of IgG4-ROD and orbital MALT lymphoma were obtained surgically, and the samples were delivered immediately to the laboratory and stored at −80 °C until the assay.

### 2.3. Microarray Analysis

Total RNA was extracted from serum and biopsy specimens using the miRNeasy Mini Kit, according to the instructions provided by the manufacturer (Qiagen GmbH, Hilden, Germany). Gene tip miRNA was extracted from the total RNA of the serum and biopsy specimens using the 3D-Gene^®^ RNA extraction reagent from a liquid sample kit (Toray Industries, Inc., Kanagawa, Japan) and was concentrated. Fluorescent labeling of RNA was hybridized to the 3D-Gene^®^ miRNA labeling kit. RNA was hybridized to a 3D-Gene^®^ Human miRNA Oligo Chip (Toray Industries, Inc.) designed to detect 2565 mature human miRNA sequences registered in miRbase release 21 (http://www.mirbase.org/), as previously described [20]. The chip was scanned using a 3D-Gene^®^ Scanner. All miRNAs with signals higher than the background signal were selected (positive call), and only miRNAs with positive call were used in subsequent analyses. Depending on the absence of stable RNA, we used the global mean normalization method (log-conversion of raw data and alignment) to normalize the expression of miRNAs in the serum and biopsy specimens [21].

### 2.4. Cluster and Principal Component Analysis

Relative expression levels of miRNAs were validated using Student’s *t*-test (*p* < 0.05). miRNAs with an expression level with a log fold change (FC) >1 or <−1 in the test sample compared with the control sample were analyzed. Cluster analysis was performed on the differentially expressed miRNAs in biopsy and serum specimens of IgG4-ROD patients, orbital MALT lymphoma patients, and healthy individuals using cluster software. Differentially expressed genes and miRNAs between any two groups of samples were identified using criteria such as *p*-value and fold change.

Principal component analysis was used to discriminate the different biological samples based on the distances of a reduced set of new variables (principal components), using the top two principal components that depicted the results in two dimensions.

### 2.5. Bioinformatics Analysis of Pathways Targeted by Differentially Expressed miRNAs

For the analysis of target genes and Kyoto Encyclopedia of Genes and Genomes (KEGG) pathways, the top 10 significantly expressed miRNAs were used. The R (3.6.2.) package clusterProfiler (version 3.16.0) [22] was used to identify the KEGG pathways of genes targeted by the differentially expressed miRNAs. A network consisting of the target genes according to the top 10 significantly different miRNAs was analyzed for the KEGG pathways. Statistical analyses were performed using the two-tailed Student’s *t*-test in R (3.6.2.). Differences with *p*-values <0.05 were considered statistically significant.

### 2.6. Bioinformatic Tools for Target Prediction, Function Enrichment Analysis, and Biomarker Identification

The R (3.6.2.) (http://www.R-project.org) packages miRNAtap (version 1.22.0; https://bioconductor.org/packages/miRNAtap/) and mygene (version 1.24.0) [23] were used to predict the target genes.

Receiver operating characteristic (ROC) curves were plotted using the pROC package (version 1.16.2) [24], which produces sensitivity and specificity metrics for each ROC curve at the optimal threshold to evaluate the predictive power of candidate miRNAs in biopsy or serum specimens of patients with IgG4-ROD and orbital MALT lymphoma. In addition, using stepwise variable selection to perform predictions using multiple miRNAs, we reduced the number of miRNAs with significant expression. Furthermore, we constructed ROC curves using a panel of miRNAs selected by the R package randomForest (version 4.6-14) and compared them with a single miRNA. A value of 1.0 indicates that the features of the model completely discriminate two diseases, whereas a value of 0.5 indicates that the features contain disease information equal to that obtained by chance alone.

## 3. Results

### 3.1. Clinical Characteristics of the Participants

The demographic and clinical features of the study population are summarized in Table 1. From these subjects, we collected serum samples from 11 patients with IgG4-ROD (mean age: 61.5 ± 17.4 years, range: 32–88 years), 13 patients with orbital MALT lymphoma (mean age: 74.2 ± 11.3 years, range: 56–89 years), and 11 healthy individuals (mean age: 66.2 ± 13.4 years, range: 38–89 years). In addition, biopsy specimens were obtained from six patients with IgG4-ROD (mean age: 69.2 ± 15.4 years, range: 43–86 years) and eight patients with orbital MALT lymphoma (mean age: 72.9 ± 10.7 years, range: 59–89 years). In three patients with IgG4-ROD, lesions were found in other organs (parotid gland, submandibular gland, and bile duct). Lesions were found in other organs (mediastinal lymph node and abdominal lymph node) in two patients with orbital MALT lymphoma. The levels of serum IgG4 were higher in patients with IgG4-ROD than in those patients with orbital MALT lymphoma.

### 3.2. MiRNA Expression Profiles in IgG4-ROD and Orbital MALT Lymphoma

Bioinformatic analysis examined 2565 miRNAs. Among the miRNAs, 287 differentially expressed miRNAs with *p* ≤ 0.05 and |log FC| > 1 were identified as potential predictors of disease type (Appendix A). As shown in Figure 1, there was a significant separation between patients with IgG4-ROD and orbital MALT lymphoma when comparing biopsy specimens, and between IgG4-ROD patients and healthy individuals when comparing serum specimens. To further narrow down miRNAs with diagnostic significance, we selected the top 10 upregulated and downregulated miRNAs of each comparison of IgG4-ROD or orbital MALT lymphoma with healthy individuals (Table 2 and Table 3). These miRNAs are presumably involved in mechanisms underlying the development of IgG4-ROD and orbital MALT lymphoma. When comparing IgG4-ROD with orbital MALT lymphoma, miR-1272 showed the highest level of upregulation, and miR-382-3p the highest level of downregulation in biopsy specimens, while miR-7974 showed the highest level of upregulation and miR-4755-3p the highest level of downregulation in biopsy specimens (Table 4 and Table 5).

In Figure 2, the heat maps of the hybridization array presents the expression of the 287 most differentially expressed miRNAs among all samples, organized into a two-way hierarchical clustering, according to miRNAs and samples. Hierarchical cluster analysis and principal component analysis of miRNAs with significant differences in expression showed that there was, to some extent, a separation between the serum samples obtained from IgG4-ROD patients or orbital MALT lymphoma, and healthy individuals (Figure 3). In IgG4-ROD patients, 237 miRNAs were significantly upregulated and 32 were significantly downregulated, compared with orbital MALT lymphoma patients and healthy individuals (Figure 4A,B). In orbital MALT lymphoma patients, 22 miRNAs were significantly upregulated and 163 were significantly downregulated, compared with healthy individuals (Figure 4C,D).

In patients with IgG4-ROD, there was no overlap of altered (upregulated and downregulated) miRNAs in biopsy or serum samples when compared with orbital MALT lymphoma patients and/or with healthy individuals (Figure 4). Five miRNAs (miR-1207-3p, miR-2355-5p, miR-3127-3p, miR-361-5p, and miR-4324) were upregulated (Figure 4A) and let-7a-5p was downregulated (Figure 4B) in the serum of IgG4-ROD patients compared with orbital MALT lymphoma patients and healthy individuals. Five miRNAs (miR-1912, miR-205-5p, miR-373-3p, miR-6770-5p, and miR-7112-3p) were downregulated in both biopsy and serum specimens of IgG4-ROD patients compared with orbital MALT lymphoma (Figure 4D), whereas there was no upregulation of any miRNA (Figure 4C). In biopsy specimens of IgG4-ROD patients, while eight miRNAs were downregulated (Figure 4A,B), a total of 103 miRNAs were upregulated compared with orbital MALT lymphoma.

In orbital MALT lymphoma patients, miR-7112-3p was downregulated in biopsy and serum specimens compared with IgG4-ROD patients and healthy individuals (Figure 4D). Analysis of serum samples showed that no miRNA was specifically upregulated in orbital MALT lymphoma patients. Analysis of biopsy specimens revealed that 106 miRNAs were upregulated, and seven miRNAs were downregulated in orbital MALT lymphoma patients compared with IgG4-ROD patients (Figure 4C,D).

Therefore, we aimed to investigate the impact of the miRNA profile on serum IgG4 level in IgG4-ROD by analyzing the expression of miRNAs and serum IgG4 level in IgG4-ROD patients. We observed a significant positive relationship between the miRNA profile and serum IgG4 and the strongest correlation was found between serum IgG4 level and miR-7854-3p expression (ρ = 0.82, *p* = 0.002) (Table 6). These results indicate that serum miRNAs were associated with serum IgG4 level.

### 3.3. Target Prediction and Enrichment of Functional Categories for Selected miRNAs in IgG4-ROD and Orbital MALT Lymphoma

Enrichment analyses of biological pathways were performed to identify the genes targeted by the top 10 upregulated and downregulated miRNAs in each group (Table 2, Table 3 and Table 4), by in silico prediction using the R (3.6.2.) package miRNAtap and mygene.

Potential pathways and genes regulated by miRNAs in patients with IgG4-ROD and orbital MALT lymphoma were estimated by bioinformatic analyses using the R package miRNAtap with clusterProfiler. The clusterProfiler was used to evaluate the enrichment of gene clusters in biological terms with a cutoff of *p* < 0.05. The complete list of KEGG pathways regulated by each group of miRNAs and the specific gene targets within the pathways for each miRNA group are shown in Figure 5. Appendix A show all the targeted genes and their corresponding targeting miRNAs in biopsy or serum specimens. In the comparison of serum samples of IgG4-ROD patients with healthy individuals, eight pathways were identified from significantly downregulated miRNAs, while 63 significantly expressed pathways (*p* < 0.05) were identified from significantly upregulated miRNAs. In the comparison of serum samples of orbital MALT lymphoma patients with healthy individuals, 25 pathways were identified from significantly downregulated miRNAs, while 21 significantly expressed pathways (*p* < 0.05) were identified from significantly upregulated miRNAs. The analysis of biopsy specimens identified 23 significantly expressed pathways (*p* < 0.05) from significantly upregulated miRNAs and 32 pathways from significantly downregulated miRNAs in IgG4-ROD patients compared with those with orbital MALT lymphoma, while 32 pathways were obtained from significantly downregulated miRNAs. In serum, eight pathways, including the KEGG pathway of genes, which was targeted by the top 10 upregulated miRNAs in patients with IgG4-ROD compared to healthy individuals (Figure 5A, Appendix A) were equally enriched. The forkhead box O (FOXO) signaling pathway was the most enriched pathway by target genes of the top 10 downregulated miRNAs in IgG4-ROD (Figure 5B, Appendix A). Comparison of orbital MALT lymphoma patients with healthy individuals showed that the Apelin signaling pathway was the KEGG pathway most enriched by target genes of the top 10 upregulated miRNAs (Figure 5C, Appendix A), and the adherens junction was the most enriched KEGG pathway for the downregulated miRNAs (Figure 5D, Appendix A). In biopsy specimens, the mitogen-activated protein kinase (MAPK) signaling pathway was the KEGG pathway most enriched by target genes of the top 10 upregulated miRNAs in IgG4-ROD patients compared with orbital MALT lymphoma (Figure 5E, Appendix A), and the longevity-regulating pathway was the most enriched pathway by the top 10 downregulated miRNAs (Figure 5F, Appendix A). The MAPK signaling pathway was the most commonly enriched KEGG pathway, except for the pathways upregulated in the serum of IgG4-ROD patients compared with healthy individuals. The transforming growth factor beta (TGFβ) signaling pathway, human papillomavirus infection, epidermal growth factor receptor tyrosine kinase inhibitor resistance, and chronic myeloid leukemia were unique pathways enriched by the target genes of downregulated miRNAs in the serum of IgG4-ROD patients.

### 3.4. Receiver Operating Characteristic (ROC) Analysis

MiR-1272 was expressed in all biopsy specimens in IgG4-ROD patients, but was not expressed in orbital MALT lymphoma patients. ROC analysis indicated perfect discrimination between IgG4-ROD and orbital MALT lymphoma patients MiR-1272, with an area under the ROC curve (AUC) of 1.0 (Figure 6A).

In serum, miR-92a-2-5p and miR-4673 were most useful for diagnosing IgG4-ROD with relatively high AUC values (0.87), while miR-7112-3p was most useful for the diagnosis of orbital MALT lymphoma with lower AUC value (0.75) (Figure 6B,C).

Subsequently, we used RandomForest to compare the levels of miRNA profile in serum samples of IgG4-ROD patients, orbital MALT lymphoma patients, and healthy individuals to verify whether serum miRNA panels selected among the 177 differentially expressed miRNAs can discriminate the two diseases. As illustrated in Figure 7, the AUC of five miRNAs (miR-1912, miR-202-3p, miR-320c, miR-361-5p, and miR-4755-3p) relevant to diagnosis was 0.92 in IgG4-ROD patients. In orbital MALT lymphoma patients, the AUC of three miRNAs (miR-4285, miR-548o-3p, and miR-7112-3p) was 0.78, with relatively high AUC values. RandomForest selection revealed that miR-202-3p was the best predictor for IgG4-ROD (95% confidence interval [CI]: 0.49-0.91; AUC area: 0.70), and miR-7112-3p was the best predictor for orbital MALT lymphoma (95% CI: 0.61-0.89; AUC: 0.75).

## 4. Discussion

In the present study, we investigated the expression of 2565 miRNAs in both biopsy and serum specimens, and identified miRNAs as biomarkers of IgG4-ROD and orbital MALT lymphoma based on the criteria of |log FC| > 1 and *p* < 0.05. In ROC analysis, miRNA panels selected by RandomForest yielded AUC of 0.92 for IgG4-ROD and 0.78 for orbital MALT lymphoma, and these values are considered to be moderately accurate for the prediction of these two diseases. This study demonstrates miRNA profiles that distinguish IgG4-ROD cases from orbital MALT lymphoma and healthy volunteers, and provides a large number of miRNAs to promote comprehensive analysis of disease status. Detection of alterations in miRNAs in biopsy and serum specimens could improve current diagnostic methods for IgG4-ROD, as these data may substantiate the evidence of disease. Statistical analysis showed that orbital MALT lymphoma had significantly decreased miR-7112-3p in the biopsy and serum specimens. These results indicate that miR-7112-3p, which is important for differentiating between IgG4-ROD and orbital MALT lymphoma.

Regulatory T (Treg) cells producing IL-10 and TGFβ contribute to enhancement of IgG4 class switch recombination and the pathogenesis of IgG4-related disease (IgG4-RD) via B cells, as well as induction of fibrosis [25]. Previous studies have confirmed the recruitment of FOXP3-positive Treg cells into the affected organs in IgG4-RD [26]. In this analysis, miR-920 targeting FOXP3 was downregulated in the serum of IgG4-ROD patients compared with healthy individuals, indicating that miR-920 may regulate the role of Treg cells in the pathogenesis of IgG4-ROD, their function and accumulation in the orbit, as well as the degree of FOXP3 expression at the site. Moreover, pathologically, both IgG4-RD and IgG4-ROD are characterized by fibrosis [27]. TGFβ produced by Treg cells contributes to fibrosis in IgG4-RD [28,29]. TGFβ regulates the expression of several miRNAs during renal fibrosis, such as miR-21, miR-29, miR-192, miR-200, and miR-433. MiR-21, miR-192, and miR-433 are positively induced by TGFβ signaling and play a pathological role in kidney diseases [30]. In contrast, both miR-29 and miR-200 families, which are inhibited by TGFβ signaling, protect the kidney from renal fibrosis [30]. In this study, we found that the top 10 upregulated miRNAs (including miR-3663-3p, miR-4673, and miR-4745-5p) and downregulated miRNAs (including miR-20b-5p, miR-6501-3p, miR-302c-5p, miR-758-5p, miR-193a-5p, miR-202-3p, let-7a-5p, and miR-379-5p) targeted 37 genes involved in the TGFβ-signaling pathway related to the pathogenesis of IgG4-ROD.

T follicular helper (Tfh) cells have also been considered important in the pathological processes of diseases, including IgG4-ROD and IgG4-RD. These cells are increased in active IgG4-RD, and the number of Tfh cells correlates with serum IgG4 and IL-4 levels [31]. B-cell lymphoma 6 (BCL6) is considered to be the master transcription factor in the functioning of Tfh cells and differentiation of naïve helper T cells to Tfh cells [32]. Strong BCL6 expression is detected in ectopic germinal centers in IgG4-RD patients [33]. In our study, we found that the miRNA targeting BCL6 was miR-9-5p, which was downregulated in biopsy specimens of IgG4-ROD patients compared with orbital MALT lymphoma. These results provide preliminary evidence that BCL6 is targeted by miR-9-5p and is implicated in IgG4-ROD. Moreover, the decrease of miR-9-5p expression in IgG4-ROD may be a potential biomarker, leading to increased expression of BCL6.

Importantly, among all the differentially expressed serum miRNAs observed in this study, the most prominent putative biomarkers of IgG4-ROD and orbital MALT lymphoma determined through machine-learning methods (randomForest) were miR-202-3p and miR-7112-3p, respectively. The miR-202-3p level is known to correlate negatively with tumor size, transcriptionally target Gli1 and inactivate the Shh signal, and has been linked to the tumorigenesis and progression of various types of human malignancies [34,35,36]. In addition, miR-202-3p inhibits gastric cancer proliferation through inducing cell apoptosis by direct interaction with Gli1 [34]. It is difficult to compare the present results with those of previous reports considering that earlier reports did not evaluate the miRNA profiles of IgG4-RD including IgG4-ROD. Hence, additional studies are warranted to establish a relationship between miR-202-3p and the cause of IgG4-ROD. Based on the evidence provided by one published report, miR-7112-3p targets the RNA-dependent protein kinase R-like endoplasmic reticulum kinase (PERK), and inhibits the function of PERK signaling, resulting in reduced apoptosis [37]. A significantly positive correlation between apoptotic and proliferative indices has been observed in gastrointestinal lymphomas of MALT [38]. In this study, the expression level of miR-7112-3p was significantly downregulated in orbital MALT lymphoma patients. Further investigations are needed to confirm the relationship between miR-7112-3p and orbital MALT lymphoma.

Pathways related to the top 10 downregulated miRNAs in the serum of IgG4-ROD patients compared with healthy individuals were mostly related to the FOXO signaling pathway, possibly due to its post-transcriptional regulation by miRNAs. Moreover, we observed downregulation of serum miRNAs that control the PI3K-AKT signaling pathway in IgG4-ROD. These signaling pathways are well-known, where the PI3K/AKT/FOXO pathway is involved in B-cell tumors, light chain recombination, receptor editing, and B-cell selection due to pathway constitutive activation [39,40]. Pathways related to the downregulated miRNAs in the serum of orbital MALT lymphoma patients compared with healthy individuals were mostly related to the adherens junction. This is a well-studied signaling pathway involved in cell adhesion of various tumors including lymphoma [41,42]. Furthermore, pathways related to miRNAs in biopsy specimens of IgG4-ROD patients compared with orbital MALT lymphoma were mostly related to the longevity-regulating pathway. To the best of our knowledge, this observation has not been reported previously.

Pathways related to the downregulated miRNAs in biopsy specimens of orbital MALT lymphoma patients compared with IgG4-ROD were mostly related to the MAPK signaling pathway. This signaling is a well-investigated pathway involved in inflammation due to regulation by the translocation gene of MALT lymphoma [43]. The analysis of pathways enriched by the target genes of significantly expressed miRNAs confirms the essential role of these transcripts and their relative targeting miRNAs in the pathogenesis of IgG4-ROD and orbital MALT lymphoma.

No unique cytological or laboratory features in peripheral blood have been identified to assist the diagnosis of orbital lymphoproliferative disorders. Therefore, diagnostic biopsy is used to define and diagnose these disorders. Recently, miRNAs have been reported to be present in plasma at detectable levels, and plasma miRNAs are more stable than mRNAs in body fluids, resistant to degradation, and easily and rapidly measurable due to their small size and stem-loop structure [44,45,46,47,48,49]. Compared with invasive biopsy, the peripheral blood-based biomarker assay is a relatively economical and noninvasive method for the detection of IgG4-ROD and orbital MALT lymphoma, owing to its easy accessibility and the low risk associated with sample collection. However, circulating miRNAs may be affected by systemic functions, such as disease complications, physiological variation, concurrent drugs and food intake. The ROC curve analysis displayed that the levels of miR-4673 and miR-92a-2-5p in serum could discriminate with greater accuracy IgG4-ROD patients from orbital MALT lymphoma patients and healthy individuals, and that miR-7112-3p could discriminate orbital MALT lymphoma patients from IgG4-ROD patients and healthy individuals. Furthermore, analysis of multiple miRNAs selected by the randomForest method showed that five miRNAs (miR-1912, miR-202-3p, miR-320c, miR-361-5p, and miR-4755-3p) in serum could more accurately discriminate IgG4-ROD patients from orbital MALT lymphoma patients and healthy individuals. Three miRNAs (miR-4285, miR-5480-3p, and miR-7112-3p) could more accurately discriminate orbital MALT lymphoma patients from IgG4-ROD patients and healthy individuals. Moreover, these miRNA panels may exhibit higher performance than single miRNAs. To confirm these preliminary results, further evaluation is warranted. Clinically, discrimination between IgG4-ROD and orbital MALT lymphoma using serum miRNAs may support clinical decision-making and allow for the timely initiation of treatment, thereby avoiding the need for biopsy.

Recently, Blosse et al. showed that four novel miRNAs (miR-150, miR-155, miR-196a, and miR-138) played important regulatory roles in the pathogenesis of gastric MALT lymphoma caused by Helicobacter pylori infection [50]. Consistent with the results reported for gastric MALT lymphoma, we found that miR-138 was expressed at higher levels in biopsy tissues of orbital MALT lymphoma patients compared with IgG4-ROD. However, similar results were not obtained for the other miRNAs. Orbital MALT lymphoma is rarely caused by Helicobacter pylori [51,52]. Therefore, the difference in miRNA profile between orbital MALT lymphoma and gastric MALT lymphoma may be attributed to the different characteristics of these two distinct types of MALT lymphoma and/or usage of different assays. Interestingly, the pathogenesis of gastric lymphoma may be associated with other risk factors, such as the hepatitis B virus, human immunodeficiency virus, Epstein–Barr virus, and human T-cell lymphotropic virus type 1 [53]. In this study, nine (miR-3192-5p, miR-3190-3p, miR-5087, miR-601, miR-650, miR-3691-5p, miR-1255b-5p, miR-205-5p, and miR-6514-5p) of the top 10 miRNAs downregulated in biopsy specimens of orbital MALT lymphoma patients were compared with IgG4-ROD targeted genes related to the KEGG pathway of human T-cell leukemia virus 1 infection. The association of human T-cell leukemia virus 1 infection with orbital MALT lymphoma has not been reported. Further investigations are required to examine such a relationship.

To date, only two studies have reported miRNA profiles of orbital malignant lymphoma, including diffuse large B-cell lymphoma [54,55]. Recently, Laban et al. showed that seven (miR-140-5p, miR-148a-3p, miR193a-5p, miR-223-3p, miR-29a-3p, miR-365a-3p, and U6 snRNA) of 399 miRNAs analyzed in a panel were upregulated in the serum of patients with non-Hodgkin lymphomas, including diffuse large B-cell lymphoma. Due to differences in the study design, the disease and control characteristics, and the assay used, their data cannot be compared with our results.

The study is limited by the retrospective design and small number of cases collected from a single institution, which may cause selection bias and confounding bias. Furthermore, IgG4 levels are variable among IgG4-ROD patients. Owing to the lack of comparable studies of IgG4-ROD including IgG4-RD, it is difficult to make a direct comparison with previous studies. A prospective study to validate the present findings and examine the clinical significance of miRNAs in different manifestations of IgG4-ROD and orbital MALT lymphoma should be performed in the future using a large clinical samples from multiple centers, with the goals of diagnosing the clinical subtypes and elucidating the pathogenesis of these diseases.

## 5. Conclusions

To the best of our knowledge, this is the first comprehensive study of the global miRNA profile of IgG4-ROD and orbital MALT lymphoma. Important miRNAs with respect to these disorders were also investigated in this study. Using this approach, we were able to identify the specific molecular pathways that may be regulated by these miRNAs. We hope that our data will contribute to the body of knowledge in this area. Further studies are warranted to confirm the present observations and clarify whether miRNA expression is related to disease recurrence and extraocular involvement. Moreover, miRNA expression may represent promising candidates in the future for the identification of disease biomarkers and the design of novel therapeutic strategies through in-depth study of orbital lymphoproliferative disorders.

## Figures and Tables

**Figure 1 jcm-09-02530-f001:**
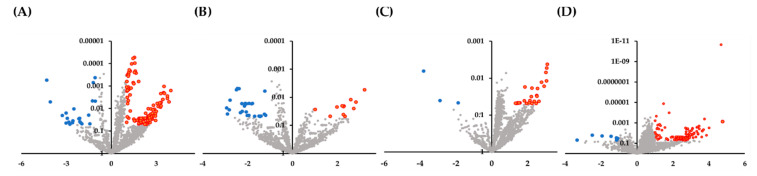
Volcano plot showing significant differentially expressed miRNAs. The *x*-axis shows the log2 fold change in miRNA expression, and the *y*-axis shows the statistical significance of the differences (*p* < 0.05). Red dots indicate significant upregulation, blue dots indicate downregulation, and grey dots indicate nonsignificant expression miRNAs. (**A**) Comparison of serum samples between the IgG4-ROD group (*n* = 11) and healthy individual group (*n* = 11). (**B**) Comparison of serum samples between the orbital MALT lymphoma group (*n* = 13) and healthy individual group (*n* = 11). (**C**) Comparison of serum samples between the IgG4-ROD group (*n* = 11) and the orbital MALT lymphoma group (*n* = 13). (**D**) Comparison of biopsy samples between the IgG4-ROD (*n* = 6) and orbital MALT lymphoma group (*n* = 8).

**Figure 2 jcm-09-02530-f002:**
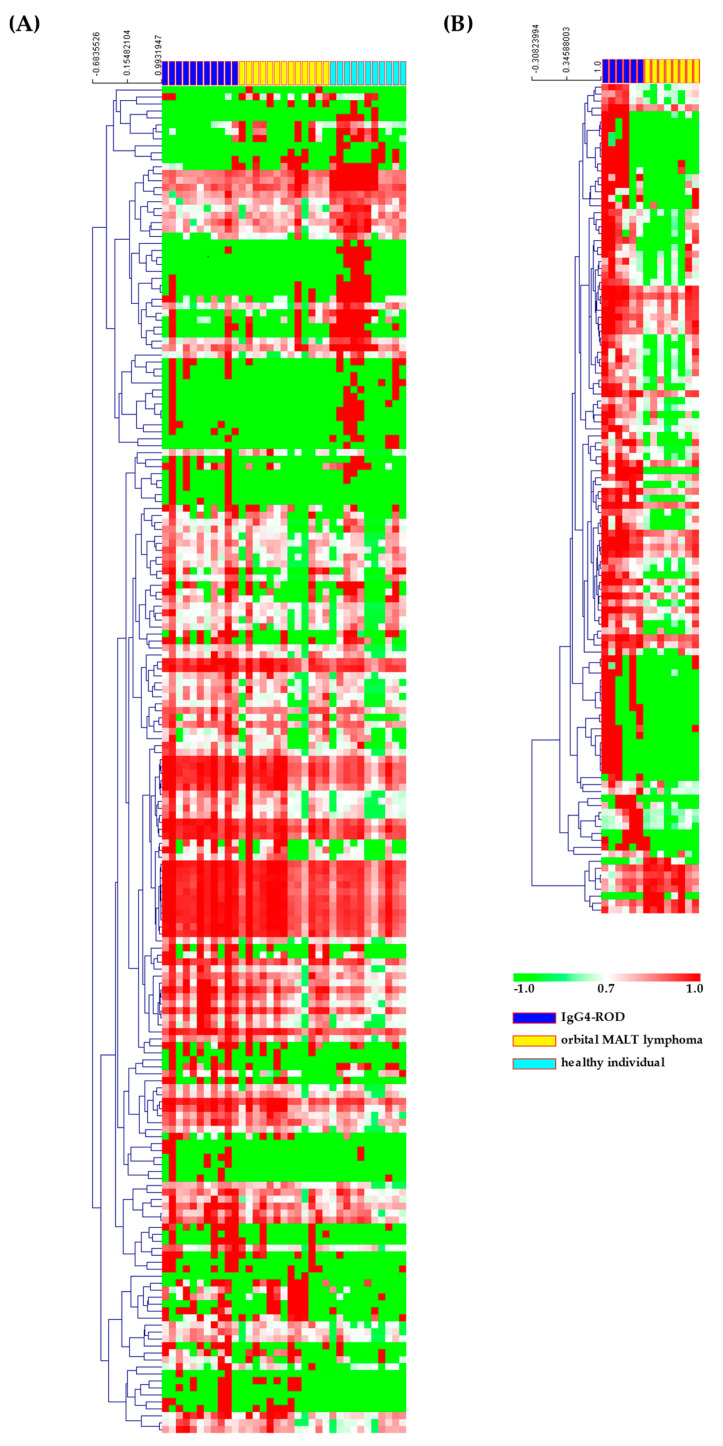
Heat maps of miRNA expression (Z-score) in IgG4-ROD, orbital MALT lymphoma and healthy individuals; colors indicate up- (red) and down-regulation (green) of gene expression. (**A**) Serum specimens; (**B**) biopsy specimens. IgG4-ROD, immunoglobulin G4-related orbital disease; orbital MALT lymphoma, orbital mucosa-associated lymph tissue lymphoma.

**Figure 3 jcm-09-02530-f003:**
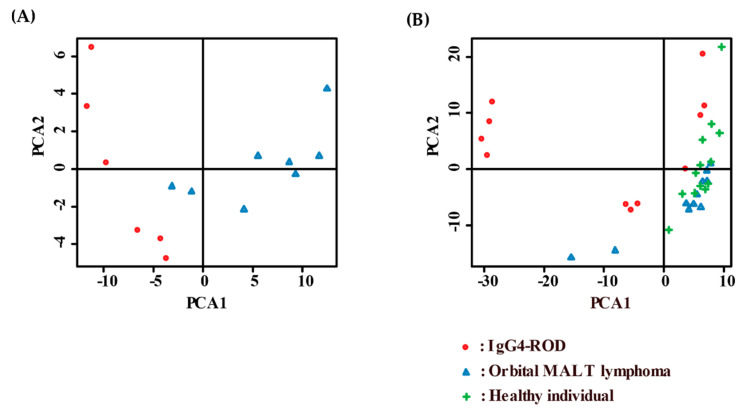
Principal component analyses of (**A**) serum specimens and (**B**) biopsy specimens. Red: IgG4-ROD, blue: orbital MALT lymphoma, green: healthy individuals. IgG4-ROD, immunoglobulin G4-related orbital disease; orbital MALT lymphoma, orbital mucosa-associated lymph tissue lymphoma.

**Figure 4 jcm-09-02530-f004:**
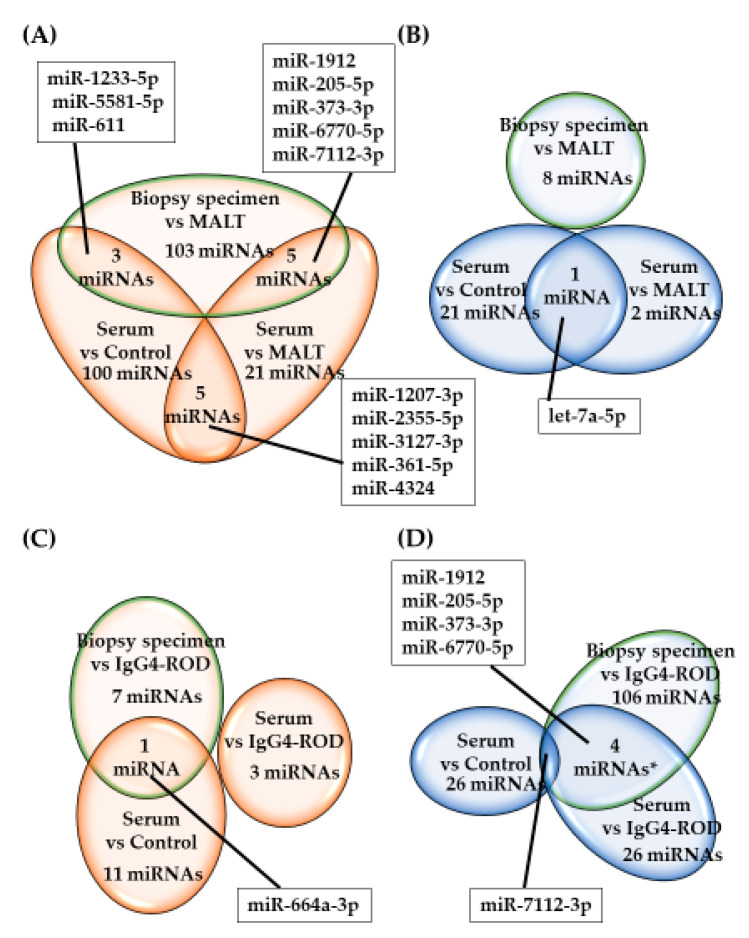
Number of miRNAs with significantly altered expression in IgG4-ROD and orbital MALT lymphoma by Venn diagram. (**A**) Upregulated miRNAs in IgG4-ROD, (**B**) downregulated miRNAs in IgG4-ROD, (**C**) upregulated miRNAs in orbital MALT lymphoma, (**D**) downregulated miRNAs in orbital MALT lymphoma. IgG4-ROD, immunoglobulin G4-related orbital disease; orbital MALT lymphoma, orbital mucosa-associated lymph tissue lymphoma.

**Figure 5 jcm-09-02530-f005:**
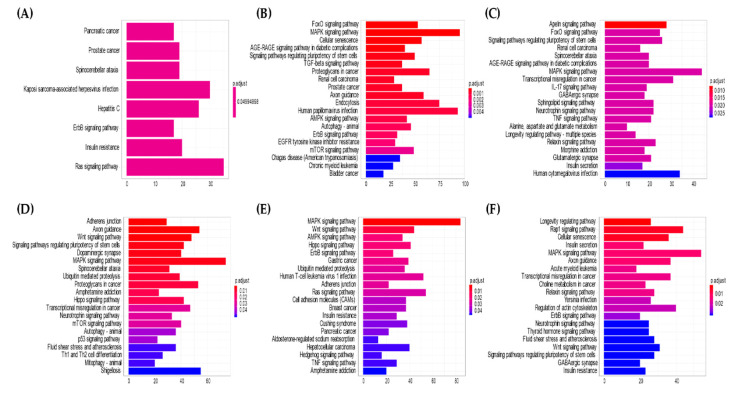
Top 20 statistically significant KEGG pathways are listed, and the colors represent *p* values of the target genes of the top 10 significantly up- and down-regulated miRNAs: (**A**) Serum miRNAs upregulated in IgG4-ROD compared with healthy individuals. (**B**) Serum miRNAs downregulated in IgG4-ROD compared with healthy individuals. (**C**) Serum miRNAs upregulated in orbital MALT lymphoma compared with healthy individuals. (**D**) Serum miRNA downregulated in orbital MALT lymphoma compared with healthy individuals. (**E**) Biopsy miRNAs upregulated in IgG4-ROD compared with orbital MALT lymphoma. (**F**) Biopsy miRNAs downregulated in IgG4-ROD compared with orbital MALT lymphoma. KEGG, Kyoto encyclopedia of genes and genomes; FoxO, forkhead box protein O; MAPK, mitogen-activated protein kinase; AGE, advanced glycation endproducts; RAGE, receptor for AGE; TGF, transforming growth factor; AMPK, accelerated mobile pages-activated protein kinase; EGFR, epidermal growth factor receptor; mTOR, mammallian target of rapamycin; IL, Interleukin; GABAergic, gamma aminobutyric acidergic; TNF, tumor necrosis factor; Th, helper T cell; Wnt, IgG4-ROD, immunoglobulin G4-related orbital disease; orbital MALT lymphoma, orbital mucosa-associated lymph tissue lymphoma.

**Figure 6 jcm-09-02530-f006:**
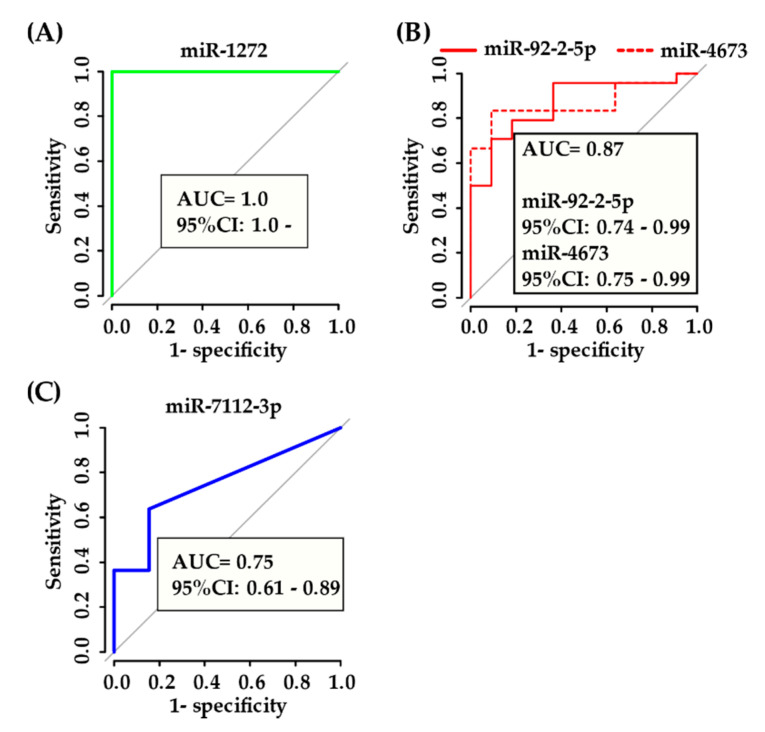
ROC curves for miRNAs with the highest AUC: (**A**) IgG4-ROD vs. orbital MALT lymphoma in biopsy specimen, (**B**) IgG4-ROD vs. orbital MALT lymphoma and healthy individuals in serum specimen, (**C**) orbital MALT lymphoma vs. IgG4-ROD and healthy individuals in serum specimen. ROC, Receiver Operating Characteristic; AUC, area under the ROC curve; CI, confidence interval; IgG4-ROD, immunoglobulin G4-related orbital disease; orbital MALT lymphoma, orbital mucosa-associated lymph tissue lymphoma.

**Figure 7 jcm-09-02530-f007:**
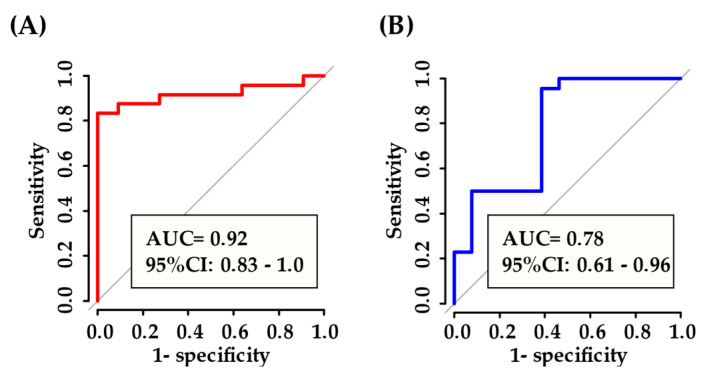
ROC curve of selected miRNAs RF: (**A**) five miRNAs (miR-1912, miR-202-3p, miR-320c, miR-361-5p, miR-4755-3p) in IgG4-ROD vs. orbital MALT lymphoma and healthy individuals in serum, (**B**) 3 miRNAs (miR-4285, miR-548o-3p, miR-7112-3p) in orbital MALT lymphoma vs. IgG4-ROD and healthy individuals in serum. ROC, Receiver Operating Characteristic; AUC, area under the ROC curve; CI, confidence interval; RF, random forest; IgG4-ROD, immunoglobulin G4-related orbital disease; orbital MALT lymphoma, orbital mucosa-associated lymph tissue lymphoma.

**Table 1 jcm-09-02530-t001:** Detailed clinicopathological and ophthalmic feature characteristics of the study participants.

	IgG4-ROD	Orbital MALT Lymphoma	Healthy Individuals
Number	17	21	11
Age (years)	64.2 ± 16.6	73.7 ± 10.9	66.2 ± 13.4
range	32–39	56–89	38–89
Female/Male	11/6	12/9	8/3
Number of other organ lesions (%)	3 (18)	2 (9)	
IgG4 in serum (mg/dL)	172.4 ± 243.2	32.4 ± 90.8	-
Ophthalmic features			
swollen eyelid	17	19	-
diplopia	3	6	-
decrease of vision	0	2	-

IgG4-ROD, immunoglobulin G4-related orbital disease; orbital MALT lymphoma, orbital mucosa-associated lymph tissue lymphoma.

**Table 2 jcm-09-02530-t002:** Top 10 upregulated and downregulated miRNAs in the serum of IgG4-ROD patients compared with healthy individuals.

Upregulated			Downregulated		
miRNA	Log Fold Change	*p*-Value	miRNA	Log Fold Change	*p*-Value
miR-550a-5p	1.54	0.00005	miR-20b-5p	−4.32	0.00054
miR-642b-3p	1.45	0.00006	miR-6501-3p	−1.09	0.00042
miR-4730	1.57	0.00009	miR-302c-5p	−1.21	0.00070
miR-4673	1.36	0.00010	miR-758-5p	−4.08	0.00498
miR-4649-5p	1.19	0.00019	miR-193a-5p	−1.26	0.00453
miR-4745-5p	1.59	0.00021	miR-920	−1.08	0.00462
miR-128-2-5p	1.27	0.00021	miR-5690	−2.5	0.01033
miR-92a-2-5p	1.54	0.00024	miR-202-3p	−3.02	0.01576
miR-3648	1.65	0.00027	let-7a-5p	−3.28	0.02033
miR-3663-3p	1.15	0.00025	miR-379-5p	−2.94	0.02703

IgG4-ROD, immunoglobulin G4-related orbital disease.

**Table 3 jcm-09-02530-t003:** Top 10 upregulated and downregulated miRNAs in the serum of orbital MALT lymphoma patients compared with healthy individuals.

Upregulated			Downregulated		
miRNA	Log Fold Change	*p*-Value	miRNA	Log Fold Change	*p*-Value
miR-518f-3p	3.21	0.00535	miR-4724-5p	−2.45	0.00481
miR-5008-3p	2.58	0.01208	miR-629-5p	−2.39	0.00489
miR-423-3p	2.82	0.01462	miR-5690	−2.5	0.00535
miR-128-3p	2.72	0.02505	miR-193a-5p	−1.23	0.00599
miR-136-3p	2.29	0.02043	miR-7112-3p	−2.79	0.01258
miR-5587-5p	2.21	0.02003	miR-6817-5p	−2.93	0.02366
miR-548o-3p	2.27	0.02136	miR-362-3p	−2.26	0.01657
miR-4655-3p	1.98	0.02211	miR-4779	−2.11	0.01631
miR-664a-3p	2.26	0.04083	miR-381-5p	−2.06	0.01679
miR-1273a	2.33	0.04706	miR-300	−2.82	0.0279

Orbital MALT lymphoma, orbital mucosa-associated lymph tissue lymphoma.

**Table 4 jcm-09-02530-t004:** Top 10 and eight upregulated and downregulated miRNAs, respectively, in biopsy specimens of IgG4-ROD patients compared with orbital MALT lymphoma patients.

Upregulated			Downregulated		
miRNA	Log Fold Change	*p*-Value	miRNA	Log Fold Change	*p*-Value
miR-1272	4.68	2.2 × 10^−11^	miR-382-3p	−2.5	0.01578
miR-3192-5p	1.48	0.00001	miR-9-5p	−3.34	0.04744
miR-3192-5p	1.79	0.0001	miR-451a	−1.95	0.01868
miR-5087	3.76	0.00041	miR-487a-3p	−1.45	0.02053
miR-601	4.77	0.00074	miR-664a-3p	−1.15	0.0285
miR-650	1.08	0.00022	miR-138-2-3p	−1.08	0.03269
miR-3691-5p	1.01	0.00061	miR-487b-3p	−1.14	0.03656
miR-1255b-5p	3.45	0.00198	miR-4783-5p	−1.14	0.04757
miR-205-5p	4.01	0.00311			
miR-6514-5p	3.38	0.00253			

IgG4-ROD, immunoglobulin G4-related orbital disease; orbital MALT lymphoma, orbital mucosa-associated lymph tissue lymphoma.

**Table 5 jcm-09-02530-t005:** Top 10 and three upregulated and downregulated miRNAs, respectively, in the serum of IgG4-ROD patients compared with orbital MALT lymphoma patients.

Upregulated			Downregulated		
miRNA	Log Fold Change	*p*-Value	miRNA	Log Fold Change	*p*-Value
miR-7974	3.1	0.00415	miR-4755-3p	−3.81	0.00639
miR-361-5p	3.07	0.00528	let-7a-5p	−2.89	0.04008
miR-205-5p	3.03	0.00695	miR-4434	−1.88	0.04621
miR-6504-5p	3.06	0.01176			
miR-7112-3p	2.77	0.01286			
miR-3655	2.92	0.01637			
miR-199a-5p	2.55	0.01893			
miR-152-5p	2.24	0.0184			
miR-6513-5p	1.85	0.01726			
miR-3127-3p	2.51	0.02914			

IgG4-ROD, immunoglobulin G4-related orbital disease; orbital MALT lymphoma, orbital mucosa-associated lymph tissue lymphoma.

**Table 6 jcm-09-02530-t006:** miRNAs are significantly related to serum IgG4; analysis of 2565 miRNAs.

miRNA	ρ	*p*-Value
miR-7854-3p	0.82	0.002
miR-6834-5p	0.81	0.003
miR-3714	0.78	0.005
miR-6748-5p	0.72	0.012
miR-6787-5p	0.72	0.013
miR-8060	0.72	0.012

IgG4-ROD, immunoglobulin G4-related orbital disease; ρ, Spearman’s rank correlation coefficient. The data show miRNAs with Spearman’s rank correlation coefficient >0.7.

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
