# Peer review of "Distinctive Tissue and Serum MicroRNA Profile of IgG4-Related Ophthalmic Disease and MALT Lymphoma"

_jcm, 2020, doi:10.3390/jcm9082530_

Round 1
Reviewer 1 Report
I commend the authors on the overall design and execution of the study. The study does address an important clinical need, which is to distinguish between IgG4-related disease and MALT lymphoma in the orbit. I agree with the authors that these clinical entities are in each other's differential diagnosis, and can be difficult to separate on clinical, radiologic and sometimes even pathologic grounds. The authors' report of diverging miRNA profiles of the two disorders is interesting, relevant and deserving of further study. It may be that miRNA profiles could be developed into a clinicopathologic test to aid in diagnostic differentiation, and in this regard it is particularly intriguing that a difference in the serum miRNA expression was also found. In addition, the authors' work raises important questions about the role of miRNAs in the pathogenesis of IgG4RD via activation of Treg and Tfh cells.
However, there are several instances where the thought process is not logically presented. This does detract significantly from the readability of the study.
Examples of sentences that need clarification:
1) Introduction, paragraph 3: "Moreover, no high-throughput miRNA expression studies, currently found to have >2,500 miRNAs, having been conducted to identify miRNAs specifically associated with the disease, which has received tremendous attention, have been performed thus far, neither global biopsied specimens nor serum miRNAs profiling in orbital lymphoproliferative disorders has been performed to date."
2) Figure 1: please make explicit the difference between what is shown in 1A and 1C
3) Figure 2: Heat map A. This would be easier to interpret if the IgG4RD and MALT patients were clustered together
4) Results 3.2 paragraph 3 on page 7: The first 2 sentences in this paragraph seem to contradict each other. "In patients with IgG4-ROD, there were no increased or decreased miRNAs in any of the biopsied specimens and serum in comparison with orbital MALT lymphoma patients and healthy individuals. Compared with orbital MALT lymphoma patients and healthy individuals, in serum, five miRNAs
J. Clin. Med. 2020, 9, x FOR PEER REVIEW 8 of 18
(i.e., miR-1207-3p, miR-2355-5p, miR-3127-3p, miR-361-5p, miR-4324) were upregulated (Fig. 4A), whereas let-7a-5p was downregulated (Fig. 4B) in patients with IgG4-ROD." Can you please clarify?
5) Discussion, paragraph 8, page 14: "The ROC curve analysis displayed that the levels of miR-4673 and miR-92a-2-5p in serum could discriminate with increased accuracy patients with IgG4-ROD from those with orbital MALT lymphoma and healthy individual, while miR-7112-3p could discriminate patients with IgG4-ROD from those with orbital MALT lymphoma and healthy individual." Since both results here are used to support discrimination between IgG4RD from MALT and healthy controls, I don't understand why they are presented as contrasting in this sentence.
I would also point out that the Methods section includes a description of how blood samples were obtained, but no description of how tissue samples were obtained and processed.
Author Response
Reviewer 1
I commend the authors on the overall design and execution of the study. The study does address an important clinical need, which is to distinguish between IgG4-related disease and MALT lymphoma in the orbit. I agree with the authors that these clinical entities are in each other's differential diagnosis, and can be difficult to separate on clinical, radiologic and sometimes even pathologic grounds. The authors' report of diverging miRNA profiles of the two disorders is interesting, relevant and deserving of further study. It may be that miRNA profiles could be developed into a clinicopathologic test to aid in diagnostic differentiation, and in this regard it is particularly intriguing that a difference in the serum miRNA expression was also found. In addition, the authors' work raises important questions about the role of miRNAs in the pathogenesis of IgG4RD via activation of Treg and Tfh cells.
Response: We are grateful to the reviewer for the encouraging comments.
However, there are several instances where the thought process is not logically presented. This does detract significantly from the readability of the study.
Examples of sentences that need clarification:
- Introduction, paragraph 3: "Moreover, no high-throughput miRNA expression studies, currently found to have >2,500 miRNAs, having been conducted to identify miRNAs specifically associated with the disease, which has received tremendous attention, have been performed thus far, neither global biopsied specimens nor serum miRNAs profiling in orbital lymphoproliferative disorders has been performed to date."
Response: We apologize for the unclear sentence. We have changed the sentence as follows. “Currently, more than 2,500 miRNAs have been identified in human genome. However, there is limited knowledge regarding the expression of miRNA. Moreover, no high-throughput miRNA expression studies have been conducted to identify miRNAs specifically associated with the disease, and no biopsy tissue and serum miRNA profiling of orbital lymphoproliferative disorders has been reported to date.’
- Figure 1: please make explicit the difference between what is shown in 1A and 1C
Response: We apologize for the error in the legend of Figure 1. We have corrected the explanations of Figure 1A-D.
- Figure 2: Heat map A. This would be easier to interpret if the IgG4RD and MALT patients were clustered together
Response: We thank the reviewer for the suggestion. We have changed heat map A as suggested.
- Results 3.2 paragraph 3 on page 7: The first 2 sentences in this paragraph seem to contradict each other. "In patients with IgG4-ROD, there were no increased or decreased miRNAs in any of the biopsied specimens and serum in comparison with orbital MALT lymphoma patients and healthy individuals. Compared with orbital MALT lymphoma patients and healthy individuals, in serum, five miRNAs (i.e., miR-1207-3p, miR-2355-5p, miR-3127-3p, miR-361-5p, miR-4324) were upregulated (Fig. 4A), whereas let-7a-5p was downregulated (Fig. 4B) in patients with IgG4-ROD." Can you please clarify?
Response: We apologize for the confusing sentence. We tried to state that in patients with IgG4-ROD, there was no overlap of altered (upregulated and downregulated) miRNAs in biopsy or serum samples when compared with orbital MALT lymphoma patients and/or with healthy individuals, as shown in Figure 4. In other words, the altered (upregulated and downregulated) serum miRNAs in IgG4-ROD compared with orbital MALT lymphoma, the altered serum miRNAs in IgG4-ROD compared with normal controls, and the altered biopsy miRNAs in IgG4-ROD compared with orbital MALT lymphoma were unique and do not overlap at all. Therefore, we have changed the sentence to:
“In patients with IgG4-ROD, there was no overlap of upregulated and downregulated miRNAs in biopsy or serum samples when compared with orbital MALT lymphoma patients and/or with healthy individuals (Fig. 4).”
- Discussion, paragraph 8, page 14: "The ROC curve analysis displayed that the levels of miR-4673 and miR-92a-2-5p in serum could discriminate with increased accuracy IgG4-ROD from those with orbital MALT lymphoma and healthy individual, while miR-7112-3p could discriminate patients with IgG4-ROD from those with orbital MALT lymphoma and healthy individual." Since both results here are used to support discrimination between IgG4RD from MALT and healthy controls, I don't understand why they are presented as contrasting in this sentence.
Response: Again we apologize for the confusing sentence. In ROC analysis of IgG4-ROD patients, we used MALT lymphoma patients and healthy individuals as controls. In ROC curve analysis of MALT lymphoma patients, we compared with IgG4-ROD patients and healthy individuals as controls. We clarify the sentence as follows:.
“The ROC curve analysis displayed that the levels of miR-4673 and miR-92a-2-5p in serum could discriminate with higher accuracy IgG4-ROD patients from orbital MALT lymphoma patients and healthy individual, and that miR-7112-3p could discriminate orbital MALT lymphoma patients from IgG4-ROD patients and healthy individuals."
- I would also point out that the Methods section includes a description of how blood samples were obtained, but no description of how tissue samples were obtained and processed.
Response: We thank the reviewer for pointing out our omission. We have added the description of how biopsied tissues were obtained and processed in Methods as follows.
“Biopsy specimens of IgG4-ROD and orbital MALT lymphoma were obtained surgically, and the samples were delivered immediately to the laboratory and stored at −80°C until assayed.”
Additional changes:
We have asked two editors of English language to check the revised manuscript before re-submission.
Reviewer 2 Report
The igG4-related ophthalmic disease is generally defined as the ocular form of immunoglobulin G4-related disease (IgG4-RD) with ocular adnexal involvement and clinicohistopathological features. For clinical, there are no biomarkers reported for IgG4-ROD diagnosis. Naoya Nezu, et al. performed microarray assay with serum and biopsied specimens from IgG4-ROD and orbital MALT lymphoma patients. They discovered several signature miRNAs for IgG4-ROD and orbital MALT lymphoma, and find the association between miRNA downregulation and MAPK signaling pathway. Overall, the data presented in this manuscript are largely convincing. I have a few suggestions:
- IgG4-ROD is a subtype of immunoglobulin G4-related disease. It will be much informative if the summary of ophthalmic manifestations is included in Table 1.
- The sample sizes for each group is quite small. Also, the variation of IgG4 level between IgG4-ROD is huge. Both compromises the observation and conclusion incorporated in this study. In the discussion section, the limitation should be magnanimously mentioned.
- The Raw data of Microarray should be deposited in public repositories, like GEO in NCBI (https://www.ncbi.nlm.nih.gov/geo/info/submission.html), and provide the access number
- The validation of miRNA changes is missing. At lease, qPCR validation on the changes of some critical miRNA, such as miR-202-3p and miR-7112-3p, are required.
Author Response
Reviewer 2
The igG4-related ophthalmic disease is generally defined as the ocular form of immunoglobulin G4-related disease (IgG4-RD) with ocular adnexal involvement and clinicohistopathological features. For clinical, there are no biomarkers reported for IgG4-ROD diagnosis. Naoya Nezu, et al. performed microarray assay with serum and biopsied specimens from IgG4-ROD and orbital MALT lymphoma patients. They discovered several signature miRNAs for IgG4-ROD and orbital MALT lymphoma, and find the association between miRNA downregulation and MAPK signaling pathway. Overall, the data presented in this manuscript are largely convincing. I have a few suggestions:
Response: We thank the reviewer for the positive comments.
1. IgG4-ROD is a subtype of immunoglobulin G4-related disease. It will be much informative if the summary of ophthalmic manifestations is included in Table 1.
Response: We thank the reviewer for the suggestion. We have added the summary of ophthalmic manifestations in Table 1.
2. The sample sizes for each group is quite small. Also, the variation of IgG4 level between IgG4-ROD is huge. Both compromises the observation and conclusion incorporated in this study. In the discussion section, the limitation should be magnanimously mentioned.
Response: We agree with the reviewer’s comments. As requested, we have added the suggested limitation to Discussion, as follows:
“The study is limited by its retrospective design and small number of cases collected from a single institution, which may cause selection bias and confounding bias. Furthermore, IgG4 levels were variable among IgG4-ROD patients.”
3. The Raw data of Microarray should be deposited in public repositories, like GEO in NCBI (https://www.ncbi.nlm.nih.gov/geo/info/submission.html), and provide the access number
Response: We thank the reviewer for the suggestion. As advised, we will deposit the raw data of microarray in GEO in NCBI
4. The validation of miRNA changes is missing. At lease, qPCR validation on the changes of some critical miRNA, such as miR-202-3p and miR-7112-3p, are required.
Response: Validation of miRNA changes is an excellent suggestion. However, since we used microarray to detect miRNA in this study, qPCR validation cannot be achieved experimentally because the detection principles of the two methods differ. The core principle behind microarray is hybridization, and this method is able to detect miRNA sequences such as variants that differ by several bases at the terminals. On the other hand, in real-time PCR, a RNA sequence that differs from the primer by even one base cannot be detected. This study does not aim to detect individual miRNAs by real-time PCR, but we intend to investigate a diagnostic method using the measurement principle of 3D-Gene® system. The microarray system we used has been reported in many previous publications (https://www.3d-gene.com/case/paper/pap_001.html), and there is no report of validation using real-time PCR that has a different detection principle. Also, the editor gave us only 5 days to respond to reviewer comments and revise the manuscript, and there is not enough time to perform real-time PCR.
Additional changes:
We have asked two editors of English language to check the revised manuscript before re-submission.